# Hypoxic mitophagy regulates mitochondrial quality and platelet activation and determines severity of I/R heart injury

Weilin Zhang[1], He Ren[2], Chunling Xu[3], Chongzhuo Zhu[1], Hao Wu[1], Dong Liu[1], Jun Wang[1], Lei Liu[1], Wei Li[4,5,6], Qi Ma[1,2], Lei Du[1], Ming Zheng[3], Chuanmao Zhang[2], Junling Liu[7], Quan Chen[1,8]*

[1]The State Key Laboratory of Membrane Biology, Institute of Zoology, Chinese Academy of Sciences, Beijing, China; [2]The Ministry of Education Key Laboratory of Cell Proliferation and Differentiation, College of Life Sciences, Peking University, Beijing, China; [3]Department of Physiology, Peking University School of Basic Medical Sciences, Peking University, Beijing, China; [4]Center for Medical Genetics, Beijing Children's Hospital, Capital Medical University, Beijing, China; [5]Beijing Pediatric Research Institute, Beijing, China; [6]MOE Key Laboratory of Major Diseases in Children, Beijing, China; [7]Department of Biochemistry and Molecular Cell Biology, Shanghai Key Laboratory of Tumor Microenvironment and Inflammation, Shanghai Jiaotong University, Shanghai, China; [8]Tianjin Key Laboratory of Protein Science, College of Life Sciences, Nankai University, Tianjin, China

*For correspondence: chenq@ioz.ac.cn

**Competing interests:** The authors declare that no competing interests exist.

**Abstract** Mitochondrial dysfunction underlies many prevalent diseases including heart disease arising from acute ischemia/reperfusion (I/R) injury. Here, we demonstrate that mitophagy, which selectively removes damaged or unwanted mitochondria, regulated mitochondrial quality and quantity *in vivo*. Hypoxia induced extensive mitochondrial degradation in a FUNDC1-dependent manner in platelets, and this was blocked by *in vivo* administration of a cell-penetrating peptide encompassing the LIR motif of FUNDC1 only in wild-type mice. Genetic ablation of Fundc1 impaired mitochondrial quality and increased mitochondrial mass in platelets and rendered the platelets insensitive to hypoxia and the peptide. Moreover, hypoxic mitophagy in platelets protected the heart from worsening of I/R injury. This represents a new mechanism of the hypoxic preconditioning effect which reduces I/R injury. Our results demonstrate a critical role of mitophagy in mitochondrial quality control and platelet activation, and suggest that manipulation of mitophagy by hypoxia or pharmacological approaches may be a novel strategy for cardioprotection.

## Introduction

Mitochondria are essential for many fundamental cellular processes, but their activities decline with age, and accumulation of dysfunctional mitochondria is associated with cardiovascular, metabolic and neurodegenerative disorders (*Wallace, 2005*; *López-Otín et al., 2013*). The crucial process of monitoring mitochondrial function and removing damaged or unwanted mitochondria is mainly achieved through mitophagy, a selective form of autophagy (*Rugarli and Langer, 2012*; *Escobar-Henriques and Langer, 2014*; *Hammerling and Gustafsson, 2014*; *Kim et al., 2007*;

*Okamoto, 2014*). Both ubiquitin- and receptor-mediated pathways mediate initiation of mitophagy (*Lazarou et al., 2015*; *Khaminets et al., 2016*; *Kirkin et al., 2009*). In response to various stresses, these pathways engage the core autophagy machinery, including ATG5, resulting in the enclosure of damaged or unwanted mitochondria in autophagosomes followed by mitochondrial degradation (*Itakura et al., 2012*; *Joo et al., 2011*; *Hara et al., 2006*). In addition to Parkin/PINK1 which mediates mitophagy in response to loss of mitochondrial membrane potential and acute stresses (*Lazarou et al., 2015*), several mitophagy receptors, including FUNDC1, localize at the outer mitochondrial membrane (*Novak et al., 2010*; *Zhang and Ney, 2010*; *Liu et al., 2012*; *Chen et al., 2014*; *Liu et al., 2014*; *Wu et al., 2014*). These receptors interact with LC3 (ATG8 in yeast) via a LIR (LC3-interacting region) motif to initiate mitophagy (*Okamoto et al., 2009*; *Kanki and Klionsky, 2010*). Distinct mitophagy pathways may be activated through transcriptional or post-translational modifications in a stress- and cellular context-dependent manner (*Durcan and Fon, 2015*; *Wei et al., 2015*; *Trausch-Azar et al., 2010*). PINK1 phosphorylates the E3 ubiquitin ligase Parkin to trigger selective autophagy of unwanted and damaged mitochondria (*Lazarou et al., 2015*). In response to hypoxia, FUNDC1 is dephosphorylated, thus enhancing the FUNDC1/LC3 interaction (*Liu et al., 2012*), while BNIP3 and NIX are transcriptionally upregulated (*Kane et al., 2014*). Ablation of NIX impairs mitochondrial clearance during red blood cell differentiation, causing a failure of erythroid maturation (*Regula et al., 2002*; *Diwan et al., 2007*; *Sandoval et al., 2008*; *Ney, 2015*). Mitochondrial distribution and morphology in Bnip3/Nix double knockout hearts were markedly perturbed (*Dorn, 2010*), as BNIP3 and NIX may sense myocardial damage in cardiomyocytes and activate pro-apoptotic mediators (*Youle and Strasser, 2008*). Parkin-mediated mitophagy is also important for adapting to acute stress (*Kubli et al., 2013*) in the myocardium, although it may not be essential for normal turnover of cardiac mitochondria (*Kubli et al., 2013*). $Pink1^{-/-}$ mice develop pathological cardiac hypertrophy (*Youle and Strasser, 2008*; *Billia et al., 2011*) accompanied by an impaired mitochondrial function in cardiomyocytes. Although these studies highlight the importance of mitophagy mediators in development and stress responses, further evidence is needed to establish the causal link between defective autophagic mitochondrial clearance and physiological outcomes under normal and disease conditions.

Circulating platelets are specialized anucleate blood cells that contain small numbers of fully functional mitochondria (*Zharikov and Shiva, 2013*). Upon stimulation, platelets release numerous cytokines and organic substances and undergo drastic morphological changes (*Gear and Camerini, 2003*; *Jackson, 2007*). Thus, platelet activation is highly energy-dependent and relies heavily on fully functional mitochondria (*Kramer et al., 2014*; *Garcia-Souza and Oliveira, 2014*). Platelet mitochondrial dysfunction leads to reduced ATP production, impaired calcium buffering, generation of reactive oxygen species (ROS) and even programmed cell death, which becomes evident in diabetes, neurodegenerative diseases and cardiovascular diseases (*Bosetti et al., 2002*; *Guo et al., 2009*; *Shrivastava et al., 2011*; *Mason et al., 2007*). It is unclear whether dysfunctional mitochondria are the cause or consequence of these diseases. General autophagy occurs in platelets in response to starvation in an ATG5-dependent fashion (*Lee et al., 2016*; *Feng et al., 2014*; *Cao et al., 2015*; *Ouseph et al., 2015*); however, the physiological relevance of autophagy in platelets is unclear. A recent report suggests that mitophagy in diabetic platelets protects against severe oxidative stress (*Lee et al., 2016*). Platelet activation is a major (patho-)physiological mechanism that underlies atherosclerosis, myocardial infarction acute coronary syndromes and ischemia/reperfusion (I/R)-related damage, and thus anti-platelet therapy is important for managing many cardiovascular diseases (*Nagareddy and Smyth, 2013*; *Gawaz et al., 2005*; *Antithrombotic Trialists' Collaboration, 2002*; *Du, 2007*; *Ruggeri, 2002*; *Li et al., 2015*). However, these diseases are still major threats to human life and new therapeutic interventions are required to protect the heart against acute I/R injury.

## Results

### Hypoxia causes extensive mitophagy in vivo in a FUNDC1-dependent manner in different tissues

To address the (patho-)physiological significance of FUNDC1-mediated mitophagy, we obtained *Fundc1* knockout embryos from the Sanger Institute (Cambridge, UK). The mice were established in

the animal center at Nanjing University and raised in the Center for Experimental Animals at the Institute of Zoology, Chinese Academy of Sciences, Beijing, China. Mice with germline knockout of *Fundc1* (F1KO) are grossly normal and fertile. They have a normal blood count and spleen size (*Figure 1—figure supplement 1*). To induce mitophagy in mice, we exposed the animals to oxygen levels of 8% for 72 hr in a hypoxic chamber. We then examined biochemical hallmarks of mitophagy by measuring mitochondrial protein levels (Tom 20 for the mitochondrial outer membrane; COXII and Tim 23 for the inner membrane), P62 levels and LC3-II expression in liver, skeletal muscle and heart from both wild-type (WT) and F1KO mice. Levels of mitochondrial proteins and P62 were reduced in response to hypoxia in tissues isolated from WT mice, although the degree of degradation differs in different tissues. Degradation of these proteins was blocked in F1KO mice. LC3-II levels were significantly increased in hypoxic wild-type tissues, whereas LC3-I levels were maintained in F1KO tissues under the same conditions (*Figure 1—figure supplement 2*).

As hypoxia affects both mitochondrial biogenesis and mitophagy in a cell context-dependent manner (*Zhu et al., 2010*; *Wu and Chen, 2015*; *Schönenberger, 2015*), we chose to examine mitophagy in platelets, because platelets have no nucleus (*Chandel, 2015*), and they are normally exposed to fluctuating oxygen levels in the circulatory system and are sensitive to hypoxic conditions. Prolonged hypoxia strongly depleted mitochondrial proteins and other mitophagy marker proteins in platelets isolated from WT but not F1KO mice (*Figure 1A*). Under similar conditions, the ER marker calnexin and the Golgi marker GM130 showed little change (*Figure 1A*). FUNDC1 is normally phosphorylated at Tyr18 by Src kinase and becomes dephosphorylated under hypoxic conditions, thus increasing its affinity with LC3 for the activation of mitophagy. We observed that FUNDC1 becomes dephosphorylated and its protein levels are decreased due to mitophagy in response to hypoxia in WT platelets (*Figure 1A*). Transmission electron microscopy also revealed a mitochondrion enclosed within a double-membrane autophagic membrane in platelets isolated from hypoxic WT mice (*Figure 1B*). However, mitophagosomes were not observed in platelets from hypoxic F1KO mice (*Figure 1B*). As expected, ex vivo assays in which platelets were isolated and then treated with hypoxia or FCCP, a commonly used inducer of mitophagy, showed almost identical mitophagy phenotypes to those in vivo (*Figure 1—figure supplement 3A,B*, *Figure 1—figure supplement 4A,B*). Next, we examined whether FUNDC1 physically interacts with LC3 to mediate hypoxia-induced mitophagy in vivo, as we previously showed in cultured cells (*Liu et al., 2012*). Co-immunoprecipitation (CO-IP) analysis revealed that FUNDC1 strongly interacted with LC3 in platelets isolated from WT mice exposed to hypoxia for 72 hr. Little interaction was detected in platelets from untreated WT mice, and no interaction was detected in the platelets from treated or untreated F1KO mice (*Figure 1C*). Similarly, ex vivo treatment of platelets exposed to hypoxia (*Figure 1—figure supplement 3C*) or FCCP (*Figure 1—figure supplement 4C*) also induced the interaction of FUNDC1 with LC3 in WT platelets, but not in F1KO platelets. Collectively, these data demonstrate that FUNDC1 interacts with LC3 to mediate mitophagy in physiological settings.

We next generated *Fundc1* platelet-specific knock-out mice using the *Pf4-Cre-LoxP/FLP* recombinant system. *Fundc1$^{fl/fl}$::Pf4$^{Cre-}$* and *Fundc1$^{fl/fl}$::Pf4$^{Cre+}$* mice were treated with hypoxia, and as illustrated in *Figure 1D,E*, platelets from *Fundc1$^{fl/fl}$::Pf4$^{Cre-}$* mice, but not *Fundc1$^{fl/fl}$::Pf4$^{Cre+}$* mice, showed an increase in LC3-II levels and a decrease in Tim23, Tom20 and P62 levels. In contrast, biochemical mitophagic hallmarks were evident in leukocytes from both *Fundc1$^{fl/fl}$::Pf4$^{Cre-}$* and *Fundc1$^{fl/fl}$::Pf4$^{Cre+}$* mice (unpublished observation). Ex vivo treatment of *Fundc1$^{fl/fl}$::Pf4$^{Cre-}$* platelets, but not *Fundc1$^{fl/fl}$::Pf4$^{Cre+}$* platelets, also induced typical mitophagic changes under hypoxia (*Figure 1—figure supplement 3D,F*). We previously showed that *Atg5*, the master regulator for general autophagy, is also required for FUNDC1-mediated mitophagy (*Liu et al., 2012*). We thus predicted that knockout of *Atg5* would also block FUNDC1-dependent mitophagy. As the deletion of *Atg5* leads to embryonic lethality in mice, *Pf4$^{Cre}$* and *Atg5$^{fl/fl}$* mice were employed to generate *Atg5$^{fl/fl}$::Pf4$^{Cre+}$* mice with platelet-specific inactivation of *Atg5*. Hypoxia failed to activate platelet mitophagy in the absence of *Atg5*, as revealed by examination of mitophagy marker proteins (*Figure 1F*, and *Figure 1—figure supplement 5*), whereas these markers were detected in cardiomyocytes and other tissues.

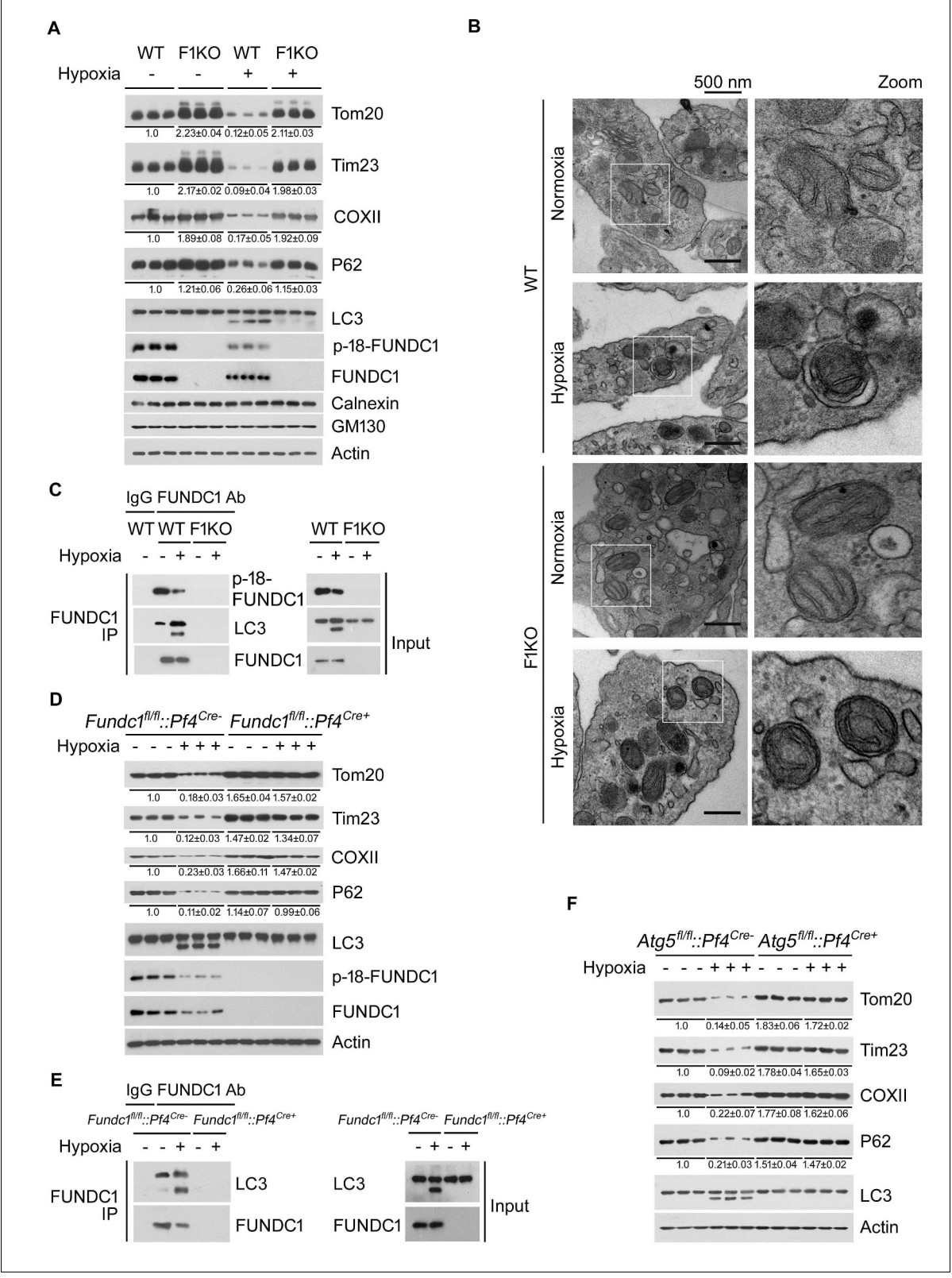

**Figure 1.** Hypoxia activates FUNDC1-dependent mitophagy in platelets *in vivo*. (**A**) F1KO mice and wild-type (WT) mice were exposed to hypoxia (8%) for 72 hr. Platelets were prepared from the treated mice and were then subjected to western blot analysis using antibodies as indicated to detect the expression levels of mitochondrial proteins and P62 (n = 28). The grayscale values of the bands were analyzed with ImageJ software and are presented below the corresponding bands to show the band intensities. (**B**) Mitophagy was detected by transmission electron microscopy. Scale bar, 500 nm.

*Figure 1 continued on next page*

*Figure 1 continued*

Quantification of mitochondria enclosed in autophagosomes was performed, and 6 out of 485 mitochondria were observed within autophagosomes in 97 platelets from WT mice exposed to hypoxia (8% $O_2$) for 72 hr. No mitochondria were detected in autophagosomes in the other groups. (C) Co-immunoprecipitation of FUNDC1 with LC3 was conducted in platelets isolated from mice treated with or without hypoxia. (D, E) Platelets were prepared from platelet-specific *Fundc1* knockout (*Fundc1$^{fl/fl}$::Pf4$^{Cre+}$*) mice and (*Fundc1$^{fl/fl}$::Pf4$^{Cre-}$*) littermates as described in (A). Mitophagy was analyzed by western blotting (D) and statistical analysis of the band intensities was performed as in (A) (n = 9). The interaction of FUNDC1 with LC3 was investigated by CO-IP (E) (n = 15). (F) Platelet-specific *Atg5* knockout (*Atg5$^{fl/fl}$::Pf4$^{Cre+}$*) mice and wild-type littermates (*Atg5$^{fl/fl}$::Pf4$^{Cre-}$*) were exposed to hypoxia and platelets were prepared from the treated mice as described in (A). Mitophagy was analyzed by western blot (n = 9).

The following figure supplements are available for figure 1:

**Figure supplement 1.** Construction and identification of F1KO mice.
**Figure supplement 2.** Detection of mitophagy in heart, liver and skeletal muscle from F1KO mice and WT mice exposed to hypoxia.
**Figure supplement 3.** *Ex vivo* analysis of platelet mitophagy induced by hypoxia.
**Figure supplement 4.** The mitochondrial toxin FCCP induces platelet mitophagy.
**Figure supplement 5.** Analysis of mitophagy in platelets with platelet-specific knockout of *Atg5*.

## FUNDC1-dependent mitophagy determines mitochondrial quality and platelet activation

Mitophagy is considered as the major mechanism to regulate mitochondrial quality. However, this has not been unambiguously demonstrated in physiological settings. We asked whether mitophagy affects the quality and functions of mitochondria in platelets. Seahorse analysis revealed that both the maximum mitochondrial oxidative capacity and the basal mitochondrial oxygen consumption rate were markedly decreased in F1KO platelets (*Figure 2A*). Consistent with our discovery that extensive mitochondrial protein degradation is prevented in F1KO platelets, we found that hypoxic treatments induced significant reduction in oxygen consumption in WT platelets, while it had little effect on platelets from F1KO mice (*Figure 2A*). Similarly, ATP production was impaired in F1KO platelets, and was also significantly reduced in hypoxia-treated WT platelets compared to untreated WT platelets (*Figure 2B*). The reduced mitochondrial activities in F1KO platelets are likely due to the accumulation of dysfunctional mitochondria owing to defective mitophagy, as mitochondrial mass and ROS were markedly increased in F1KO platelets whereas mitochondrial membrane potential was significantly decreased (*Figure 2C–E*). These data demonstrate that FUNDC1-mediated mitophagy determines the quality and functional integrity of mitochondria in platelets.

Platelet activation involves drastic structural changes, a process that demands high levels of energy and fully functional mitochondria (*Bosetti et al., 2002*). We thus predicted that mitophagy has functional consequences for platelet activation. Exposure of WT mice to hypoxia significantly reduced platelet aggregation, platelet spreading and surface expression of P-selectin upon treatment with thrombin (*Figure 2F,H,I*), ADP and collagen (unpublished observations). In contrast, aggregation of platelets from F1KO mice was markedly reduced in response to thrombin (*Figure 2F,H,I*). Similar phenotypes were observed in platelets from both *Fundc1$^{fl/fl}$::Pf4$^{Cre-}$* mice (*Figure 2J,L,M*) and *Atg5$^{fl/fl}$::Pf4$^{Cre-}$* mice (*Figure 2N,P,Q*). Mitochondrial activity, represented by mitochondrial oxygen consumption rates, was closely related to the level of platelet aggregation, with a correlation coefficient close to one in platelets from WT, *Fundc1$^{fl/fl}$::Pf4$^{Cre-}$* and *Atg5$^{fl/fl}$::Pf4$^{Cre-}$* mice (*Figure 2G,K,O*). In contrast, platelet activation was compromised in F1KO platelets, *Fundc1$^{fl/fl}$::Pf4$^{Cre+}$* and *Atg5$^{fl/fl}$::Pf4$^{Cre+}$* platelets, and was not further reduced by hypoxia (*Figure 2*). This reason for this is that genetic ablation of *Fundc1* or *Atg5* significantly blocked mitophagy, leading to accumulation of damaged mitochondria. These mitochondria are compromised, but still function at a reasonably high level. This contrasts with platelets in wild-type mice under prolonged hypoxia, where mitochondria are largely depleted and have lost most of their functions. This is also a specific effect, because there was no significant change in the levels of the ADP receptor P2Y12 and the thrombin receptors (PARs), and downstream signaling mediated by the platelet integrin receptor

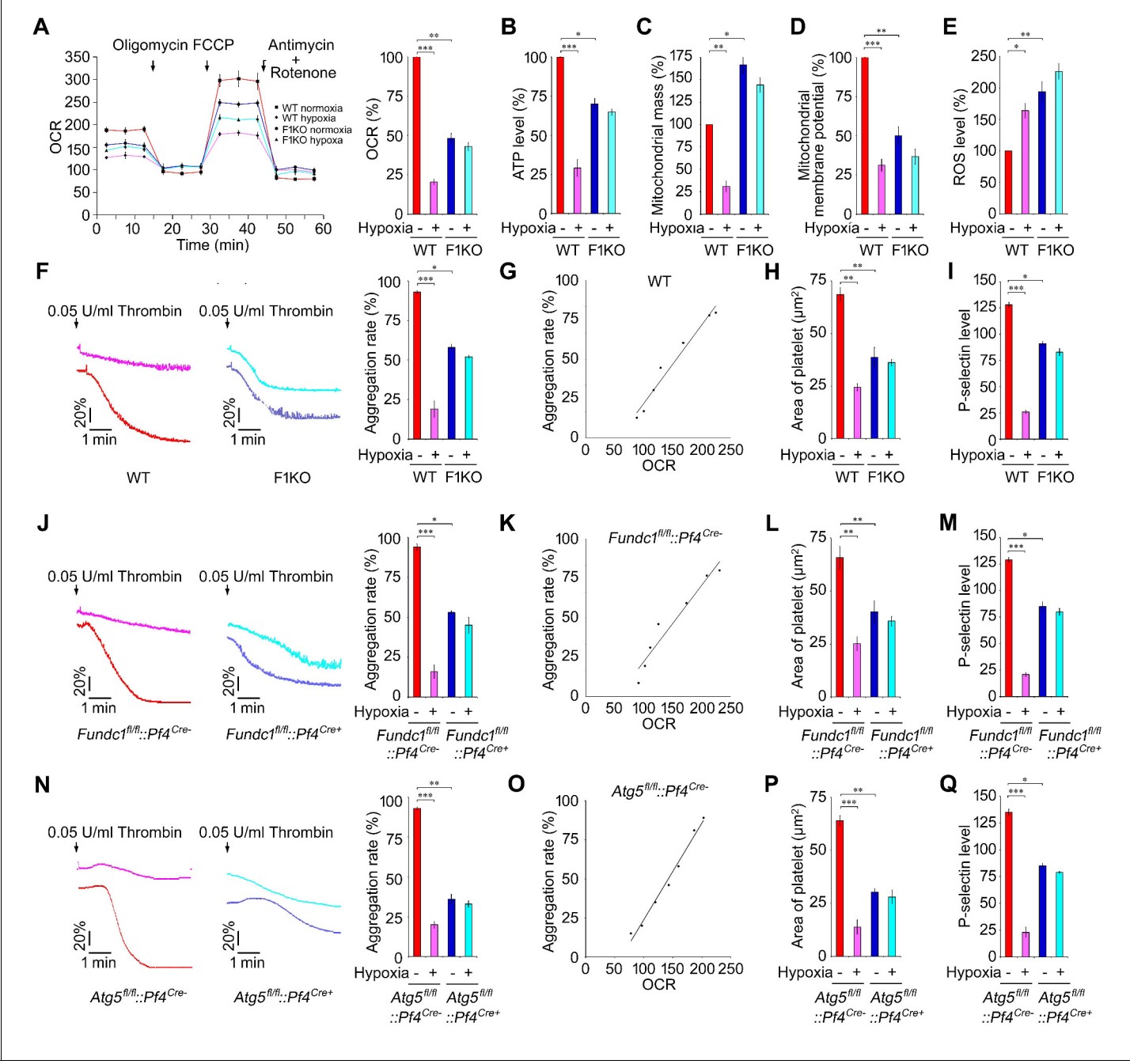

**Figure 2.** Hypoxic mitophagy determines mitochondrial quality and platelet activation *in vivo*. F1KO mice and wild-type (WT) mice (**A–I**), *Fundc1*<sup>fl/fl</sup>*::Pf4*<sup>Cre-</sup> and *Fundc1*<sup>fl/fl</sup>*::Pf4*<sup>Cre+</sup> mice (**J–M**) and *Atg5*<sup>fl/fl</sup>*::Pf4*<sup>Cre-</sup> and *Atg5*<sup>fl/fl</sup>*::Pf4*<sup>Cre+</sup> mice (**N–Q**) were treated for 72 hr under 8% hypoxia. (**A**). The oxygen consumption rate (OCR) of platelets was measured by Seahorse (**A**, left) and was normalized to the level in normoxic WT platelets (**A**, right). (**B**) ATP level was detected and normalized to the percentage in normoxic WT platelets. (**C**) Mitochondrial mass was measured by flow cytometry after staining with NAO (20 nM) and was normalized to the percentage in normoxic WT platelets. (**D**) Mitochondrial membrane potential was analyzed by flow cytometry and normalized to the percentage in normoxic WT platelets. (**E**) ROS levels were detected by flow cytometry. DCF fluorescence intensity was quantified using BD FACS Calibur and CellQuest software. ROS levels were normalized to the percentage in normoxic WT platelets. (**F, J, N**) Platelet aggregation was analyzed (n = 16) using a turbidometric aggregometer. (**G, K, O**) The correlation between mitochondrial OCR and aggregation was analyzed in platelets treated with different concentrations of thrombin (0.00625, 0.0125, 0.025, 0.05, 0.10, 0.20, 0.40 U/mL). (**H, L, P**) Platelet spreading was analyzed after treatment with 0.05 U/ml thrombin. At least 96 platelets were analyzed in each group. (**I, M, Q**) P-selectin expression on platelets after activation with 0.05 U/ml thrombin was analyzed by flow cytometry. Data from three separate experiments are presented as mean ± s.e.m. *p<0.05. **p<0.01. ***p<0.001.

*Figure 2 continued on next page*

*Figure 2 continued*

The following figure supplements are available for figure 2:

**Figure supplement 1.** FUNDC1 deficiency has no effect on megakaryocyte production or the expression of platelet membrane receptors.

**Figure supplement 2.** FUNDC1 deficiency does not affect platelet apoptosis induced by ABT737.

IIbIIIa (*Figure 2—figure supplement 1B–G*). Megakaryocyte differentiation was also identical in WT and F1KO mice (*Figure 2—figure supplement 1A*). Deficiency of *Fundc1* did not enhance apoptosis in platelets in response to ABT737 (10 µM) under our experimental conditions, as measured by phosphatidylserine (PS) exposure and activation of caspase-3 (*Figure 2—figure supplement 2*).

## A cell-penetrating peptide prevents hypoxia-induced mitochondrial dysfunctions and platelet inactivation

We previously used synthetic cell-penetrating peptides containing the LIR motif of FUNDC1 to show that the unphosphorylated LIR motif, but not the LIR motif phosphorylated at Tyr18, can block the interaction of FUNDC1 with LC3, thus preventing mitophagy in cultured cells (*Liu et al., 2012*). Intraperitoneal injection of the unphosphorylated peptide (P), but not the phosphorylated peptide which was used as a control (C), effectively blocked hypoxia-induced mitochondrial protein and P62 degradation and the conversion of LC3-I into LC3-II in WT platelets (*Figure 3A*). Neither peptide had any effect in F1KO platelets (*Figure 3A*). Treatment of platelets with the peptides alone did not induce mitophagic phenotypes (unpublished observations), although fluorescent labeling analysis revealed that these peptides were able to penetrate into platelets to exert their anti-mitophagic effects (*Figure 3—figure supplement 1*). CO-IP results demonstrated that peptide P, but not peptide C, inhibited the FUNDC1/LC3 interaction in WT platelets under hypoxic conditions (*Figure 3B*). Importantly, peptide P, but not C, effectively blocked the hypoxia-induced decrease in oxygen consumption rate and ATP generation in WT platelets (*Figure 3C–D*). Also, peptide P, but not C, prevented the loss of mitochondrial mass, loss of mitochondrial membrane potential and the increase of ROS in hypoxic WT platelets (*Figure 3F,G*). In contrast, neither peptide had a significant effect on mitochondrial activities in F1KO platelets under hypoxic conditions (*Figure 3C–G*).

Next, we examined the effects of peptides on the functional consequences of platelet activation. Thrombin, the commonly used platelet activator, was used to treat platelets isolated from WT or F1KO mice with or without hypoxic exposure. As expected, peptide P, but not C, significantly prevented the reduction of platelet aggregation, platelet spreading and surface expression of P-selectin upon exposure to hypoxia for 72 hr in WT platelets (*Figure 3I–K*). This was due to the inhibition of mitophagy by P, but not C, as confirmed by western blot analysis (*Figure 3H*). In contrast, activation of platelets from F1KO mice was markedly reduced in response to thrombin under normoxic conditions, due to the reduced but sustained level of mitochondrial functions (*Figure 3I–K*). The thrombin-induced activation of platelets from F1KO mice was not further reduced by hypoxia, or rescued by peptide treatments, because mitophagy was inhibited in F1KO platelets (*Figure 3I–K*). Similar phenotypes were observed in platelets from *Fundc1*<sup>fl/fl</sup>::*Pf4*<sup>Cre-</sup> and *Fundc1*<sup>fl/fl</sup>::*Pf4*<sup>Cre+</sup> mice (*Figure 3L–O*).

## Mitophagy in platelets protects the heart from I/R injury

Myocardial ischemia and reperfusion initiate intracellular changes that cause progressive cardiac injury, including arrhythmias, myocardial stunning, no-reflow and cell death, and platelets play important roles in the pathogenesis of acute myocardial infarction following I/R (*Yellon and Hausenloy, 2007*; *Gawaz, 2004*). We thus examined the role of mitochondrial functional integrity in platelets on I/R-induced heart injury. To avoid the complication of I/R-induced mitophagy in cardiomyocytes, we used *Fundc1*<sup>fl/fl</sup>::*Pf4*<sup>Cre+</sup> mice and applied a widely used model of myocardial infarction by occluding the left coronary artery for 0.5 hr followed by reperfusion for 24 hr (*Figure 4A*). *Fundc1*<sup>fl/fl</sup>::*Pf4*<sup>Cre+</sup> mice had much smaller heart infarct size (IF/AAR) than *Fundc1*<sup>fl/fl</sup>::*Pf4*<sup>Cre-</sup> mice as revealed by the 2, 3, 5 - triphenyltetrazolium chloride (TTC) staining assay, and there was no

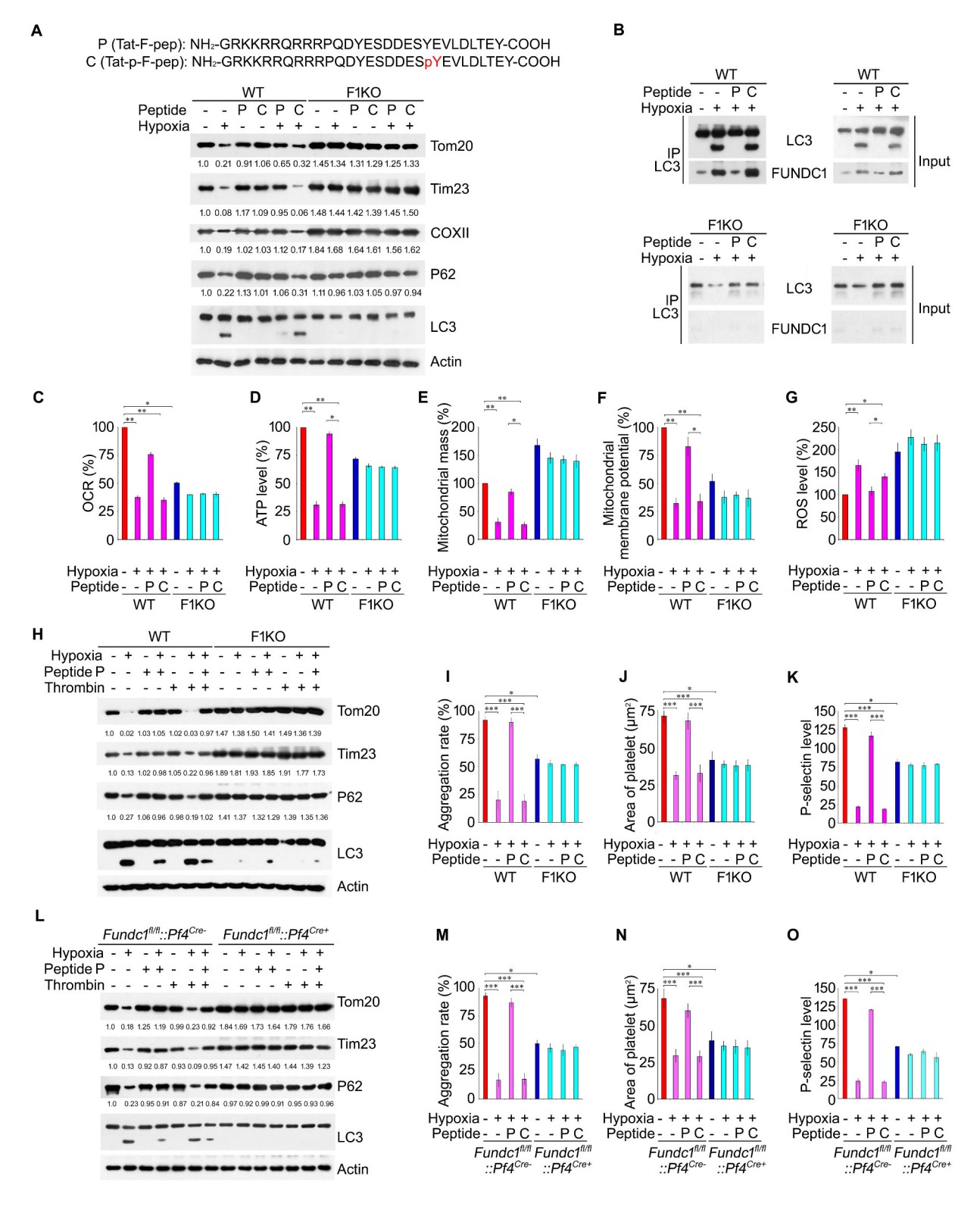

**Figure 3.** A cell-penetrating peptide prevents hypoxia-induced mitochondrial dysfunctions and platelet inactivation. F1KO mice and wild-type (WT) mice (A–K), and *Fundc1*[fl/fl]*::Pf4*[Cre-] and *Fundc1*[fl/fl]*::Pf4*[Cre+] (L–O) were treated with a cell-penetrating peptide mimicking the dephosphorylated (P) and phosphorylated (C) LIR domain of FUNDC1 every 24 hr at 1 mg/kg for 72 hr under 8% hypoxia. Platelet mitophagy was analyzed by western blot using antibodies against mitochondrial proteins and P62 (A). The grayscale values of the bands were analyzed with ImageJ software and are presented below

*Figure 3 continued on next page*

*Figure 3 continued*

the corresponding bands to show the band intensities. The FUNDC1/LC3 interaction was examined by CO-IP (**B**). Platelet OCR (**C**), ATP level (**D**), mitochondrial mass (**E**), mitochondrial membrane potential (**F**) and ROS levels (**G**) were measured as described in *Figure 2*. (**H, L**) Platelet mitophagy was examined by western blot and statistical analysis was performed as in (**A**). Platelet aggregation (**I, M**), platelet spreading (**J, N**) and P-selectin expression (**K, O**) were analyzed (n = 16) as described in *Figure 2*. Data from three separate experiments are presented as mean ± s.e.m. *p<0.05. **p<0.01. ***p<0.001.

The following figure supplement is available for figure 3:

**Figure supplement 1.** Detection of FITC-labeled peptides by flow cytometry.

significant difference in area at risk/left ventricle (AAR/LV) (*Figure 4B–D*). Echocardiography analysis showed that myocardial functions in *Fundc1^{fl/fl}::Pf4^{Cre+}* mice are normal under normoxic conditions, because the basal levels of fractional shortening (FS) and ejection fraction (EF) are comparable to those in *Fundc1^{fl/fl}::Pf4^{Cre-}* mice (*Figure 4E–G*). This suggests that specific knockout of *Fundc1* in platelets does not affect heart functions. Following I/R, *Fundc1^{fl/fl}::Pf4^{Cre-}* mice showed a significant reduction in myocardial functions, as measured by FS and EF, compared to *Fundc1^{fl/fl}::Pf4^{Cre+}* mice (*Figure 4E–G*). Similar phenotypes were observed in *Atg5^{fl/fl}::Pf4^{Cre-}* and *Atg5^{fl/fl}::Pf4^{Cre+}* mice (*Figure 4—figure supplement 1*).

Initially, I/R-induced-platelet activation and release of platelet-derived mediators can exacerbate tissue injury and result in no-reflow after reperfusion due to microembolism following ischemia and reperfusion (*Gawaz, 2004*). We further predict that early hyperactivation of platelets causes severe heart injury and hypoxic conditions in blood due to malfunction of the circulation, which may in turn affect mitochondrial function in platelets. To test this, we first detected surface expression of P-selectin on platelets in affected hearts by immunohistochemistry at different time points following I/R (*Figure 4A,H*). In agreement with early platelet activation as reported previously, we found that P-selectin surface expression was induced following I/R in hearts. Also, consistent with the high degree of heart injury, P-selectin surface expression was significantly higher in *Fundc1^{fl/fl}::Pf4^{Cre-}* mice than that in *Fundc1^{fl/fl}::Pf4^{Cre+}* mice during I/R (*Figure 4H,M*). I/R injury causes malfunction of the heart which reduces the oxygen supply in the blood and the reduced oxygen level may trigger mitophagy in platelets. Indeed, blood oxygen saturation measurements revealed that I/R treatment significantly reduced the oxygen saturation levels (*Figure 4I*). Western blotting analysis revealed that platelet mitophagy occurred in *Fundc1^{fl/fl}::Pf4^{Cre-}* mice within 24 hr of I/R, and it was blocked in *Fundc1^{fl/fl}::Pf4^{Cre+}* mice (*Figure 4J*). Correspondingly, mitochondrial OCR and ATP levels were both significantly reduced in platelets from *Fundc1^{fl/fl}::Pf4^{Cre-}* mice after I/R (*Figure 4K,L*). Thus, we detected two phases of platelet activation depending on the oxygen levels. Following initial ischemia and reperfusion, the heart injury causes platelet activation at the injury site and in the microcirculation, which causes heart malfunctions and reduced blood oxygen levels. At later time points, both the surface expression level of P-selectin and the aggregation level of platelets isolated from *Fundc1^{fl/fl}::Pf4^{Cre-}* progressively declined compared to sham controls as the reperfusion time increased (*Figure 4M,N*). These results further support our notion that mitophagy occurs in platelets in response to decreased oxygen levels in platelets to regulate mitochondrial activity and platelet activation.

## Hypoxic mitophagy in platelets preconditions mice against I/R-induced myocardial infarction

Hypoxic preconditioning was previously shown to effectively prevent I/R-induced cardiac injury (*Yellon and Hausenloy, 2007*; *Gawaz, 2004*; *Cadenas et al., 2010*; *Gottlieb and Mentzer, 2013*), although the exact protective mechanisms are not fully understood. We hypothesized that hypoxic mitophagy in platelets may be important for the protective effect of hypoxic preconditioning against I/R-induced cardiac injury. To test this, we exposed *Fundc1^{fl/fl}::Pf4^{Cre-}* mice to hypoxia for 72 hr in the presence or absence of peptides before I/R was performed (*Figure 5A*). Hypoxic preconditioning of the *Fundc1^{fl/fl}::Pf4^{Cre-}* mice strongly blocked I/R-induced heart injury (*Figure 5B,C*), and this was reversed by the administration of peptide P, but not the control peptide C. Likewise, hypoxic preconditioning maintained heart functions, as evaluated by FS and EF, and this protective effect

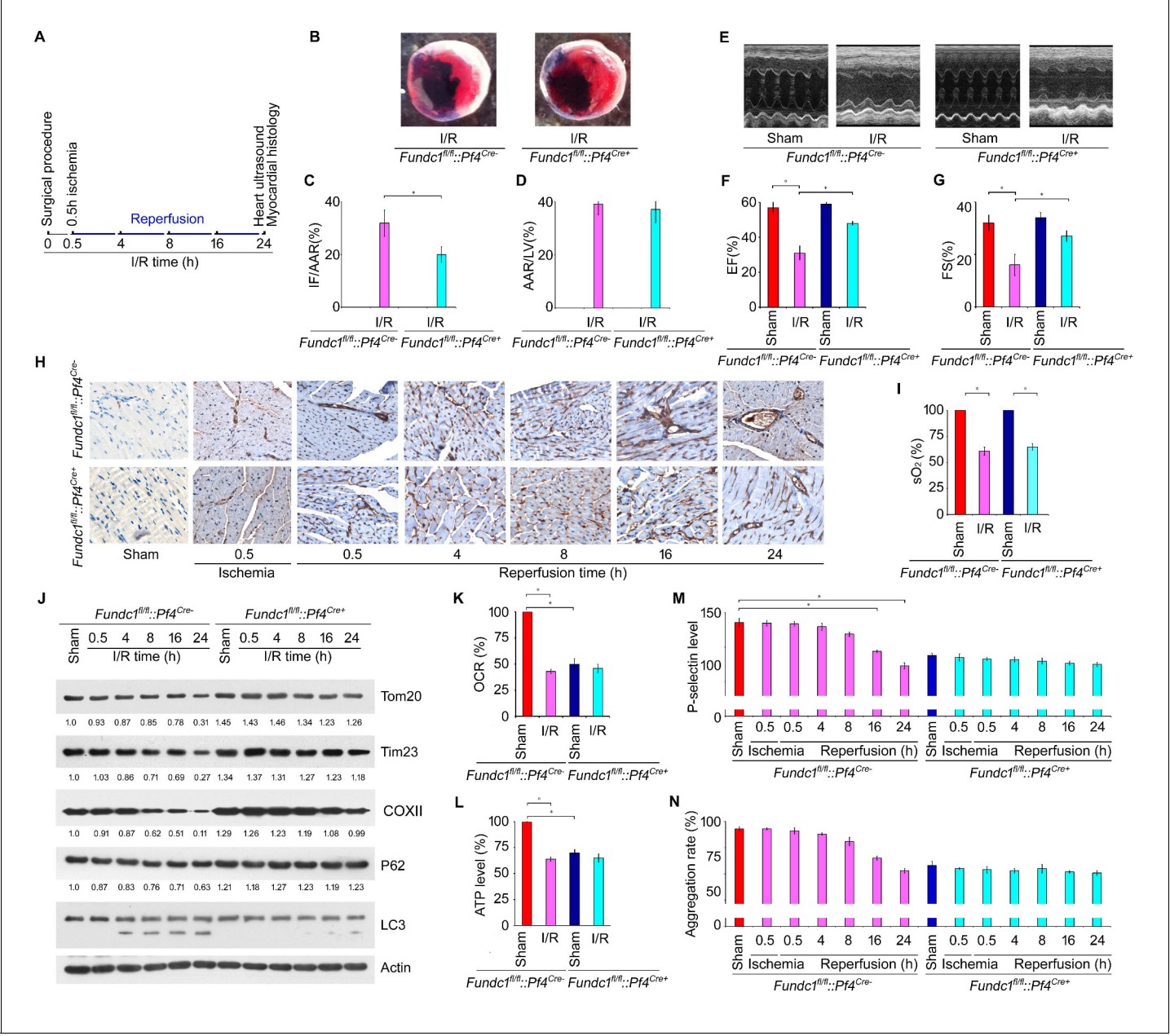

**Figure 4.** Mitophagy in platelets protects the heart from excessive ischemic/reperfusion (I/R) injury. (**A**) Timetable and procedure of ischemia/reperfusion (I/R) experiments. (**B–D**) Representative sections and quantitative analysis of the area at risk (AAR) and infarct size (IF) of hearts from *Fundc1$^{fl/fl}$::Pf4$^{Cre-}$* and *Fundc1$^{fl/fl}$::Pf4$^{Cre+}$* I/R mice (0.5 hr ischemia/24 hr reperfusion). Representative images of hearts (**B**) and quantitative data for infarct size (IF) and area at risk (AAR) are shown (**C, D**). n = 13 mice in each group. Left ventricle: LV. (**E–G**) Fractional shortening (FS) and ejection fraction (EF) were analyzed by echocardiography in *Fundc1$^{fl/fl}$::Pf4$^{Cre-}$* and *Fundc1$^{fl/fl}$::Pf4$^{Cre+}$* I/R mice (0.5 hr ischemia/24 hr reperfusion). n = 13 mice in each group. (**H**) P-selectin surface expression was analyzed by immunohistochemistry at the indicated time points. (**I**) Oxygen saturation measurements of blood from the LV reveal that I/R treatment significantly reduced the blood oxygen saturation (0.5 hr ischemia/24 hr reperfusion). (**J**) Western blot analysis of mitophagy in platelets from I/R mice at different time points of reperfusion. Typical western blot bands are shown. The expression levels of mitochondrial proteins and P62 were detected and the grayscale values of the bands were analyzed with ImageJ software. The values are presented below the corresponding bands to show the intensities. (**K**) Oxygen consumption rate (OCR) of platelets from mice treated as in (**A**) was measured by the standard Seahorse protocol (0.5 hr ischemia/24 hr reperfusion). Mitochondrial OCR was quantitatively analyzed following the manufacturer's instructions and normalized to platelet number. (**L**) ATP production was detected with an ATP determination kit from Life Technologies according to the manufacturer's instructions, then normalized to the percentage in sham-treated *Fundc1$^{fl/fl}$::Pf4$^{Cre-}$* platelets (0.5 hr ischemia/24 hr reperfusion). (**M**) P-selectin surface expression was analyzed by flow cytometry (0.5 hr ischemia/24 hr reperfusion). (**N**) Aggregation of platelets from mice treated as in

*Figure 4 continued on next page*

*Figure 4 continued*

(**A**) was induced by α-thrombin (0.05 U/ml) (0.5 hr ischemia/24 hr reperfusion). Quantitative data from three separate experiments are shown as mean ± s.e.m. *p<0.05.
The following figure supplement is available for figure 4:

**Figure supplement 1.** Conditional knockout of *Atg5* in platelets reduces I/R injury.

was reduced by peptide P, but not peptide C (*Figure 5D,E*). This is probably because the combination of hypoxia and peptide regulates the level of mitophagy in platelets, which determines the degree of platelet activation. Indeed, hypoxic preconditioning prevented P-selectin surface expression in platelets under I/R (*Figure 5F*), and this was reversed by peptide P, but not peptide C. Similarly, both surface expression of P-selectin and platelet aggregation were reduced by I/R or hypoxia or both, and these changes were prevented by peptide P, but not peptide C (*Figure 5G,H*). Moreover, peptide P, but not C, also prevented the hypoxia-induced loss of mitochondrial functions as measured by mitochondrial OCR and ATP levels in platelets from *Fundc1$^{fl/fl}$::Pf4$^{Cre-}$* mice 24 hr after I/R (*Figure 5I,J*). Biochemical analysis revealed that peptide P, but not C, prevented hypoxic and I/R-induced mitophagy (*Figure 5K*). Collectively, these data demonstrated that hypoxic mitophagy in platelets is a key component of the protective effect of hypoxic preconditioning on I/R-induced heart injury.

## Discussion

Here we demonstrated that mitophagy regulates mitochondrial quality control and mediates mitochondrial degradation in (patho-)physiological settings. We also showed that hypoxic mitophagy dictates the level of mitochondrial activity, which is related to the level of platelet activity. In response to hypoxia caused by either environmental stress conditions or by prolonged I/R, mitophagy occurs in vivo in a FUNDC1- and ATG5-dependent manner, and a peptide containing the unphosphorylated LIR motif of FUNDC1 effectively blocks hypoxia- and I/R-induced mitophagy in platelets (*Figure 3*, *Figure 4*). This suggests a unique strategy for intervening in mitophagy in vivo. Administration of the unphosphorylated peptide prevents extensive mitophagy under prolonged hypoxic conditions, thus blocking mitochondrial depletion and platelet inactivation. In contrast, loss of FUNDC1 results in compromised mitochondrial functions and reduces platelet activity to a low but constant level which is insensitive to hypoxia or peptides. These results show that hypoxic mitophagy is functionally significant for mitochondrial quality control and degradation and suggest that a new layer of regulation is involved in the response of platelets to (patho-)physiological stimuli.

Several mitophagy mediators including PINK1/Parkin, NIX/BNIP3, FUNDC1 and Bcl-w have been described to regulate mitophagy (*Lazarou et al., 2015*; *Novak et al., 2010*; *Zhang and Ney, 2010*; *Liu et al., 2012*; *Liu and Shio, 2008*). Elegant studies have focused on the molecular process by which the autophagy machinery is engaged to activate mitochondrial clearance in cultured cells (*Lazarou et al., 2015*; *Novak et al., 2010*; *Liu et al., 2012*). Ablation of the genes encoding mitophagy receptors causes the accumulation of abnormal mitochondria in different tissues (*Novak et al., 2010*; *Zhang and Ney, 2010*; *Gong et al., 2015*). However, definitive proof for mitophagic regulation of mitochondrial quality *in vivo* is still lacking. For example, Parkin is rare in normal hearts and is dispensable for constitutive mitophagic quality control. Parkin-mediated mitophagy acts as an acute stress-reactive pathway to remove damaged mitochondria as originally proposed (*Dorn, 2016*; *Youle and Narendra, 2011*). Taking advantage of the unique platelet system, we clearly showed that FUNDC1-mediated mitophagy regulates mitochondrial quality and quantity in both normoxic and in hypoxic conditions. It appears that platelets in FUNDC1 knockout mice have increased mitochondrial mass (*Figure 2*), decreased mitochondrial function (*Figure 2*) and reduced platelet activation (*Figure 3*) even under normoxic conditions, rendering the platelets insensitive to peptide P and hypoxic treatments. We also detected biochemical hallmarks of mitophagy in tissues such as liver, muscle and heart in wild-type mice when they were treated with hypoxia. although the level of change differs in different tissues (*Figure 1—figure supplement 2*). However, these changes are largely blocked in *Fundc1*-deficient mice. These data suggest that there is a tissue-specific effect of

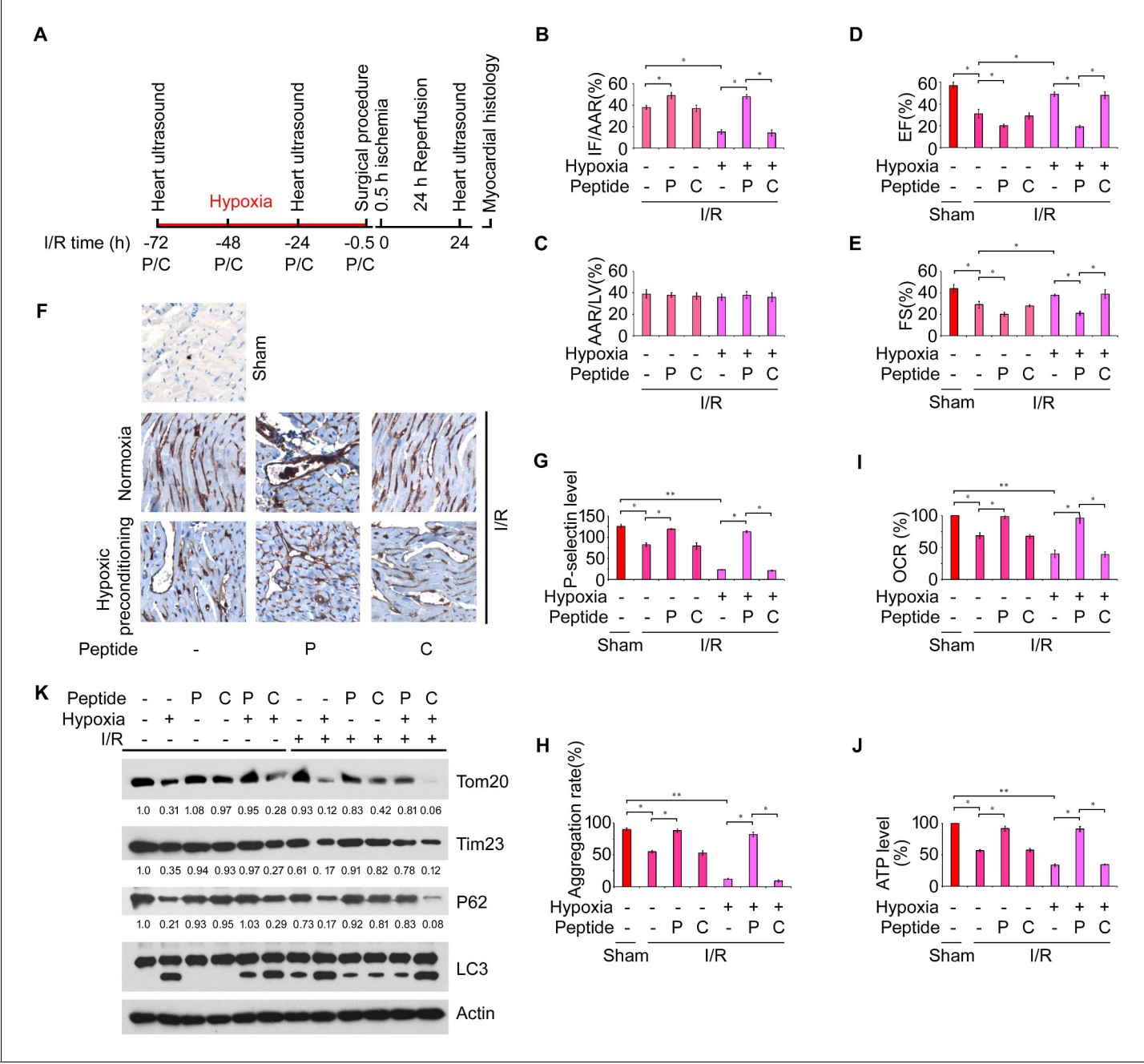

**Figure 5.** Hypoxic mitophagy in platelets preconditions mice against I/R-induced myocardial infarction and heart failure. (A) Timetable and procedure of I/R experiments with hypoxic preconditioning. *Fundc1^fl/fl^::Pf4^Cre-^* mice were pre-treated with hypoxia (8%) for 72 hr, and then subjected to I/R. (B–C) Quantitative results for area at risk (AAR) and infarct size (IF) of hearts from treated mice. n = 15 mice in each group. Left ventricle: LV. (D–E) Fractional shortening (FS) and ejection fraction (EF) were analyzed by echocardiography. n = 15 mice in each group. (F, G) P-selectin surface expression was analyzed by immunohistochemistry (F) and flow cytometry (G). Aggregation of platelets (H), OCR (I), ATP level (J) and platelet mitophagy (K) were analyzed as above. In (K), the expression levels of mitochondrial proteins and P62 were detected and the grayscale values of the bands were analyzed with ImageJ software. The values are presented below the corresponding bands to show the intensities. Quantitative data from at least three separate experiments are shown as mean ± s.e.m. *p<0.05. **p<0.01.

mitophagy in response to hypoxia, and this is likely due to distinct mechanisms for balancing mitophagy and mitochondrial biogenesis in the hypoxic stress response. It is also possible that different mitophagy mediators are involved in different tissues and further investigation is underway to study mitophagy in *Parkin* and *Nix* knockout mice.

Mitochondria are a major target in hypoxic/ischemic injury and play critical functions in the response to hypoxia, ischemia and reperfusion (*Nanayakkara et al., 2015*; *Ham and Raju, 2016*). Perturbation of mitochondrial homeostasis and cellular energetics worsens the outcome following hypoxic-ischemic insults (*Nanayakkara et al., 2015*; *García-Rivas and Torre-Amione, 2009*). Selective removal of damaged mitochondria by mitophagy was thus regarded as a critical protective mechanism against I/R injury in the heart and other organs. It is well documented that Parkin-mediated mitophagy is an important regulatory mechanism in myocardial ischemia reperfusion, myocardial infarction and heart failure (*Huang et al., 2011*; *Gong et al., 2015*). Platelets have been implicated in these processes (*Gawaz, 2004*), but the underlying mechanism remains elusive. It is interesting to note that platelet mitophagy has dual roles in platelet activation and in I/R injury. In the acute phase, mitophagy maintains mitochondrial quality and function by getting rid of damaged mitochondria that are 'toxic', so as to support platelet activation. This leads to occlusion, reduced blood oxygen levels and myocardial infarction, and explains how loss of mitophagy reduces I/R injury to a certain extent. In the prolonged phase, extensive elimination of mitochondria (maybe even good-quality and partially functional ones) due to hypoxia renders the platelets less active, preventing excess I/R-induced injury. This may serve as a negative self-regulation mechanism to prevent excessive platelet activation and myocardial injury, which has long been observed in the clinical setting. We thus revealed that hypoxic mitophagy in platelets is causally linked with I/R-induced heart injury, and that mitophagy in platelets plays a dual role in the I/R injury of hearts.

We have shown that hypoxic preconditioning induces extensive mitochondrial degradation and drastic inactivation of platelet activity, thus preventing I/R injury. Our results may help to explain how preconditioning with ischemia/hypoxia or clinically available volatile anesthetics such as isoflurane could have a beneficial effect in reducing myocardial I/R injury (*Fullmer et al., 2013*; *Liu et al., 2015*). These preconditioning agents affect oxygen levels in the blood or in the heart (*Fullmer et al., 2013*; *Liu et al., 2015*), thus activating mitophagy in platelets to reduce platelet activation and I/R injury. This is of clinical significance and our preliminary clinical studies have shown that mitophagy in platelets occurs in patients with hypoxemia and myocardial infarction (unpublished observations). We also showed that peptide P can effectively block hypoxia-induced mitophagy. Administration of peptide P is the first plausible method for manipulating mitophagy in vivo and fighting IR-induced heart injury. In contrast to previous interventions that center on cardiomyocytes or platelet surface receptors (*Zhang et al., 2016*; *Gawaz, 2004*; *McFadyen and Jackson, 2013*; *Du, 2007*), our results suggest a novel strategy for cardioprotective intervention by targeting mitophagy in platelets through hypoxia and administration of agents that promote mitophagy. The detrimental effects of acute I/R injury contribute to morbidity and mortality through a wide range of pathologies including myocardial infarction, ischemic stroke, acute kidney injury and circulatory arrest-related damage, and I/R injury is also a major challenge during organ transplantation and vascular and general surgery (*Yellon and Hausenloy, 2007*). Therefore, targeting mitophagy both in cardiomyocytes (*Dorn et al., 2015*) and platelets (*Figure 5*) or reducing mitochondrial activity by pharmacological and biochemical means will offer new life-saving strategies for fighting these deadly clinical complications.

## Materials and methods

### Reagents, antibodies

von Willebrand factor (VWF) was prepared and purified from cryoprecipitates according to the method described previously (*Yuan et al., 2009*). α-thrombin was purchased from Enzyme Research Laboratories (1801 Commerce Drive, South Bend, IN 46628). Dimethyl sulfoxide (DMSO) and bovine serum albumin (BSA) were bought from Sigma (Sigma Corporation of America, St. Louis, Missouri). Goat anti-rabbit immunoglobulin conjugated with horseradish peroxidase (GAR-HRP), goat anti-mouse immunoglobulin conjugated with horseradish peroxidase (GAM-HRP), FITC-conjugated goat anti-rabbit IgG (FITC-GAR) and FITC-conjugated goat anti-mouse IgG (FITC-GAM) were bought

from DAKO (2966 Industrial Row Troy, Michigan 48084). The following antibodies were employed in the present investigation: anti-Tim23 (1:1,000, BD Biosciences, Becton,Dickinson and Company, New Jersey, USA. RRID:AB_398755), anti-Tom20 (1:1,1000, Abcam, Cambridge, CB4 0FL, UK), anti-P62 (1:1,000, BD Biosciences, Becton, Dickinson and Company, New Jersey, USA), anti-LC3B polyclonal antibody (1:1,000, BD Biosciences, Becton,Dickinson and Company, New Jersey, USA) and anti-actin (1:1,000, Sigma Corporation of America, St. Louis, Missouri. RRID:AB_1844539). Anti-p-FUNDC1 (Tyr 18) (1:500) and anti-FUNDC1 (1:1,1000) polyclonal antibodies were produced by immunizing rabbits with synthesized and purified phosphorylated and non-phosphorylated peptides from FUNDC1 (Abgent, SuZhou, China). The cell-penetrating peptides P (NH2-GRKKRRQRRRPQD YESDDESYEVLDLTEY-COOH) and C (NH2-GRKKRRQRRRPQDYESDDEpSYEVLDLTEY-COOH, pS indicates phosphorylated serine) were synthesized by Shanghai Qiangyao Company (Shanghai, China). All the above peptides were HPLC purified and had a purity of greater than 95%.

## Animals

Mice in this study were maintained in the Center for Experimental Animals at the Institute of Zoology, Chinese Academy of Sciences, Beijing, China. The animal research was approved by the Animal Care and Use Committee of the Institute of Zoology, Chinese Academy of Sciences. Adult C57BL/6 (B6) mice (MGI:1856709) were treated in a hypoxia chamber and exposed to hypoxia (8% oxygen) or normoxia for 72 hr. Only male mice were used in this study. The mice were randomly allocated to each treatment group. No exclusion or noninclusion parameters were employed in this study. Although the researchers were not blinded to the experimental status of the animals, no subjective judgment was conducted. All the procedures involving mice conformed to the Directions for the Care and Use of Laboratory Animals and followed the protocols approved by the Committee for Animal Research of the Institute of Zoology, Chinese Academy of Sciences, Beijing, China.

*Fundc1* knock-out embryos (F1KO) were from the Sanger Institute (Cambridge, UK) and the mice, with a C57BL/6 genetic background, were established in the animal center at Nanjing University and bred in the animal center of the Institute of Zoology, Chinese Academy of Sciences, Beijing, China. The following primers were used for PCR identification of the mice: Primer 1, 5'-CCAACTGACC TTGGGCAAGAACAT-3'; Primer 2, 5'-GTATGCTCAGGAGGTATAGGCTGAC-3'; Primer 3, 5'-CACACCTCCCCCTGAACCTGAAA-3'; Primer 4, 5'-CCTCATAAACATTCTTGCTAGCAA-3'.

To specifically knock out *Fundc1* in platelets, *Fundc1*-floxed mice (*Fundc1$^{fl/fl}$::Pf4$^{Cre-}$*) were crossed with *Pf4$^{Cre}$* mice to obtain *Fundc1$^{fl/wt}$::Pf4$^{Cre+}$* mice. Further mating produced *Fundc1$^{fl/fl}$:: Pf4$^{Cre+}$* mice with platelet-specific deficiency of *Fundc1*. The following primers were used for PCR identification of the mice: Primer 1, 5'-GTATGCTCAGGAGGTATAGGCTGAC-3'; Primer 2, 5'-CC TCATAAACATTCTTGCTAGCAA-3'; Primer 3, 5'-CCCATACAGCACACCTTTTG-3'; Primer 4, 5'-TGCACAGTCAGCAGGTT-3'.

To specifically delete *Atg5* in platelets, *Atg5*-floxed mice (*Atg5$^{fl/fl}$*) were crossed with *Pf4$^{Cre}$* mice to obtain *Atg5$^{fl/wt}$::Pf4$^{Cre+}$* mice. Further mating produced *Atg5$^{fl/fl}$::Pf4$^{Cre+}$* mice with deletion of *Atg5* in platelets. PCR was employed to genotype the mice, and deficiency of *Atg5* in platelets was further confirmed by western blot. The following primers were used for PCR identification: Primer 1, 5'-GAATATGAAGGCACACCCCTGAAATG-3'; Primer 2, 5'-GTACTGCATAATGGTTTAACTCTTGC-3'; Primer 3, 5'-ACAACGTCGAGCACAGCTGCGCAAGG-3'; Primer 4, 5'-CAGGGAATGGTGTC TCCCAC-3'; Primer 5, 5'-AGGTTCGTTCACTCATGGA-3'; Primer 6, 5'-TCGACCAGTTTAGTTACCC-3'.

For in vivo assays, mice were treated intraperitoneally (i.p.) with the cell-penetrating peptides P or C (1 mg/kg).

## Platelet preparation

For the experiments involving experimental animals and human subjects, approvals were obtained from the Institutional Review Board of the Institute of Zoology, Chinese Academy of Sciences. A collection of fresh blood and isolation of washed mouse platelets were conducted as described previously (*Lee et al., 2016*; *Feng et al., 2014*; *Ouseph et al., 2015*). Briefly, the *Fundc1$^{fl/fl}$::Pf4$^{Cre+}$* mice (8 to 10 weeks old) and their *Fundc1$^{fl/fl}$::Pf4$^{Cre-}$* littermates were anesthetized with pentobarbital (70 mg/kg, i.p.). Whole blood from the abdominal aorta was harvested into acid-citrate-phosphate (ACD) solution, and platelet rich platelets (PRP) was obtained by centrifuging at 180 g for 10

min. Then the PRP was diluted in ACD solution containing apyrase (1 U/ml) and centrifuged at 800 g at room temperature for 10 min. The harvested platelet pellets were resuspended in the Tyrode's buffer containing 5 mM HEPES, 5.5 mM glucose, 1 mM $MgCl_2$, 0.34 mM $Na_2HPO_4$, 137 mM NaCl, 12 mM $NaHCO_3$, 2 mM KCl and 0.35% bovine serum albumin (BSA).

In the studies involving platelets from I/R models and sham controls, blood was collected from the left ventricle.

## Platelet aggregation

For investigations of platelet aggregation in vitro, washed platelets ($3 \times 10^8$/ml) were pre-treated with P (10 µM), C (10 µM), or hypoxia or vehicle control at room temperature for different lengths of time, and then the aggregation of platelets was stimulated with α-thrombin (0.05 U/ml). The pre-treated platelets were then analyzed using a turbidometric aggregometer for platelets (Precil LBY-NJ, Xinpusen, Beijing, China) with a stirring speed of 1000 revolutions per minute (rpm) at a temperature of 37°C.

## Western blots

Platelets were prepared as described previously (*Lee et al., 2016*; *Feng et al., 2014*; *Ouseph et al., 2015*). Washed mouse platelets ($3 \times 10^8$/ml) resuspended in modified tyrode buffer (MTB) were lysed in 2 × platelet lysis buffer (1 mM dithiothreitol, 0.15 M NaCl, 0.1 M Tris, 2% Triton X-100 and 0.01 M EGTA, pH 7.4) containing E64 (0.1 mM), 1/100 aprotinin and phenylmethylsulfonyl fluoride (PMSF, 1 mM), at a volume ratio of 1:1 on ice for 0.5 hr. The platelet lysates were subjected to analysis by sodium dodecyl sulfate polyacrylamide gel electrophoresis (SDS-PAGE) and western blot using antibodies including anti-Tim23, anti-Tom20, anti-P62, anti-FUNDC1, anti-p-18-FUNDC1 and anti-LC3. Actin served as a loading control to ensure that each protein input was similar. In ex vivo assay, the separated platelets were incubated with FCCP or hypoxia (2% oxygen) at RT for 120 min. The treated platelet samples were subjected to SDS-PAGE and western blot analysis. The concentration of vehicle control (DMSO) in each platelet sample was no more than 0.05%.

## Coimmunoprecipitation of FUNDC1 with LC3

The platelet lysates were pre-treated with anti-FUNDC1 antibody at 4°C for 120 min, and then they were further incubated with protein G beads at 4°C overnight. Bead-bound platelet proteins were extracted using SDS sample buffer and subjected to western blotting analysis with the antibodies indicated in the Figures.

## Flow cytometry

To determine the platelet mitochondrial inner transmembrane potential ($\Delta\Psi$m), the pre-treated mice platelets were incubated with TMRE to a final concentration of 100 nM. The pre-treated platelet samples were further incubated with TMRE at 37°C in the dark for 20 min and $\Delta\Psi$m was examined by flow cytometry.

Reactive oxygen species (ROS) were detected in the pre-treated platelets using 6-carboxy-2',7'-dichlorodihydrofluorescein diacetate (DCFH-DA) according to the manufacturer's instructions (Beyotime, China). Briefly, the pre-treated mouse platelets were further incubated with 10 µM DCFH-DA in the dark at 37°C for 15 min, and the platelets were washed with MTB three times. The platelets were then treated with FCCP (5 µM) or vehicle at RT for 120 min or hypoxia (2% oxygen) for 120 min. The DCF fluorescence intensity of the platelets was examined by flow cytometry.

For detection of P-selectin surface expression, pre-treated mice platelets were further treated with anti-P-selectin antibody or control mouse IgG at RT for 0.5 hr, and then incubated with FITC-GAM at RT in the dark for 0.5 hr. The platelets were then subjected to flow cytometry analysis.

## Confocal microscopy

Pre-treated mouse platelets ($3 \times 10^8$/ml) were prepared as described above. Lab-Tek (Nunc) cover-slips were pre-incubated with 30 µg/ml VWF at 4°C overnight and blocked with 5% BSA at RT for 60 min. The pre-treated platelet samples were additionally stimulated by thrombin (0.05 U/ml) and allowed to adhere to the VWF-coated cover slips at 37°C as described previously. The adherent platelet samples were further fixed with 4% paraformaldehyde and blocked with 5% BSA. The

platelets were treated with phalloidin (10 µg/ml) at RT for 2 hr, then subjected to confocal micros-copy (Lab-Tek MicroImaging, Inc. LSM510) (63 × objective lens).

## Metabolic assay

The oxygen consumption rate (OCR) was measured under basal conditions, and in the presence of 0.25 µM oligomycin, 5 µM fluoro-carbonyl cyanide phenylhydrazone (FCCP), 1 µM rotenone and antimycin. The ATP level was examined in platelet lysates using an ATP Bioluminescence Assay Kit HS II (Roche) according to the manufacturer's directions as described previously (*Dorn et al., 2015*).

## Electron microscopy analysis

Platelets isolated from mice were fixed in 2.5% glutaraldehyde for 2 hr at room temperature. Elec-tron microscopy analysis was conducted as described previously (*Liu et al., 2012*) by employing a 120 kV Jeol electron microscope at 80 kV.

## Ischemia/reperfusion mouse model

Adult *Fundc1^{fl/fl}::Pf4^{Cre-}* and *Fundc1^{fl/fl}::Pf4^{Cre+}* mice (C57BL/6, 8–10 weeks old) were subjected to the ischemia/reperfusion procedure as described previously (*Zhang et al., 2016*). The assessments of the infarct size and area-at-risk were conducted as described previously. Briefly, the mice were anesthetized with 70 mg/kg pentobarbital and then ventilated on a Harvard rodent respirator through a tracheostomy. Midline sternotomy was conducted and snare occlusion was performed in the neighborhood of the left anterior descending artery. Myocardial ischemia/reperfusion surgery was performed by tightening the reversible snare for 0.5 hr and then releasing it for reperfusion for 24 hr. After reperfusion for 24 hr, the mice were sacrificed and the hearts were collected for meas-urements of infarct size and analysis of mitophagy.

## Measurement of infarct size

To analyze infarct size, the mice were anesthetized with 100 mg/kg pentobarbital, and then the hearts were excised and the ascending aortas were catheterized from the distal Valsalva to the sinus of Valsalva. The area-at-risk was detected by retrograde perfusion with alcian blue (0.05%). The coro-nary arteries were occluded and the occlusion site was perfused with 0.05% alcian blue. Finally, the hearts were collected and frozen at −80°C for 10 min. Each frozen heart was cut into 4–7 slices, which were incubated in sodium phosphate buffer with 2,3,5-triphenyl-tetrazolium chloride (1%) to detect the infarct. The area of the left ventricle and the area of the infarct were measured using the planimetry method with Image J software as described previously (*Zhang et al., 2016*). The infarct size was analyzed by dividing the infarct area by the area-at-risk.

## Echocardiography

Echocardiography was conducted as described previously (*Zhang et al., 2016*). Briefly, the mice were anesthetized with 70 mg/kg pentobarbital and then subjected to echocardiography analysis. Cardiac contractile functions were analyzed by assessing fractional shortening and ejection fraction, and the systolic left ventricular internal diameter was measured to demonstrate the left ventricular dilation.

## Immunohistochemistry of heart tissues

Hearts of I/R mice and sham controls were analyzed by immunohistochemistry as described previ-ously (*Zhang et al., 2016*). Briefly, the hearts were fixed in paraformaldehyde (4%) in PBS (pH 7.4) immediately and then embedded with paraffin. Paraffin-embedded heart sections were then rehy-drated and treated with hydrogen peroxide (1%). After being washed gently with PBS, the heart sec-tions were blocked with serum (10%). A rabbit antibody (1:2000) against P-selectin was used in the immunostaining before the application of the secondary antibodies. The heart sections were then visualized and analyzed by microscope.

## Statistical analysis

No statistical method was employed to predetermine the size of the samples. Statistical analysis was conducted using SPSS 18.0 software (SPSS Inc, RRID:SCR_002865) and GraphPad PRISM version

5.01 (GraphPad Software, Inc. RRID:SCR_002798). Data are shown as mean ± s.e.m. The datasets were tested for normality of distribution using the Kolmogorov-Smirnov test. The two-sided unpaired Student's $t$-test was performed to compare two data groups with normal distribution. Non-parametric data were analyzed using the Mann-Whitney $U$-test. One-way ANOVA with Bonferroni *post hoc* analysis was employed for multiple group comparisons. *$p<0.05$. **$p<0.01$. ***$p<0.001$.

## Acknowledgements

We are grateful to all laboratory members for useful discussions. We wish to thank Professors Ruiping Xiao, David Huang and Dr Xudong Liao for their kind suggestions about the experiments and critical comments on the manuscript. We wish to thank Noboru Mizushima from Department of Biochemistry and Molecular Biology, Graduate School and Faculty of Medicine, The University of Tokyo for providing ATG5-floxed mice (ATG$^{f/f}$). We wish to thank Dr. Yingchun Hu, Guopeng Wang and Junlin Teng for their technical advice and support with EM. This research was supported the Natural Science Foundation of China (31520103904), the National Key Research and Development Program (2016YFA0500201, 2016YFA0100503), Special Fund for Strategic Pilot Technology Chinese Academy of Sciences (QYZDJ-SSW-SMC004) and the Beijing Natural Science Foundation of China (5161002) to QC, the Natural Science Foundation of China (31301130) and China postdoctoral grant (2013 M541041) to WZ and the Natural Science Foundation of China (31201042, 31471306) to LL.

## Additional information

### Funding

| Funder | Grant reference number | Author |
| --- | --- | --- |
| China Postdoctoral Grant | 2013M541041 | Weilin Zhang |
| National Natural Science Foundation of China | 31301130 | Weilin Zhang |
| National Natural Science Foundation of China | 31520103904 | Quan Chen |

The funders had no role in study design, data collection and interpretation, or the decision to submit the work for publication.

### Author ORCIDs

Wei Li, http://orcid.org/0000-0001-7430-6019
Quan Chen, http://orcid.org/0000-0001-7539-8728

### Ethics

Animal experimentation: This study was performed in strict accordance with the recommendations in the Guide for the Care and Use of Laboratory Animals of the Institute of Zoology, Chinese Academy of Sciences. All of the animals were handled according to approved institutional animal care and use committee (IACUC) protocols (#08-133) of the Institute of Zoology, Chinese Academy of Sciences. The protocol was approved by the Committee on the Ethics of Animal Experiments of the Institute of Zoology, Chinese Academy of Sciences (Permit Number:2014-31301130). All surgery was performed under sodium pentobarbital anesthesia, and every effort was made to minimize suffering.

### Author contributions

WZ, Conception and design, Acquisition of data, Analysis and interpretation of data, Drafting or revising the article, Contributed unpublished essential data or reagents; HR, CX, CZh, HW, DL, JW, LD, Conception and design, Acquisition of data, Drafting or revising the article; LL, CZha, Conception and design, Acquisition of data, Analysis and interpretation of data, Drafting or revising the article; WL, Conception and design, Analysis and interpretation of data, Drafting or revising the article; QM, MZ, Analysis and interpretation of data, Drafting or revising the article, Contributed unpublished essential data or reagents; JL, QC, Conception and design, Analysis and interpretation of data, Drafting or revising the article, Contributed unpublished essential data or reagents

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
