## [Decision Letter]

Thank you for submitting your article "Mitochondrial quality regulates platelet activation and determines the severity of ischemia/reperfusion heart injury" for consideration by *eLife*. Your article has been favorably evaluated by Tony Hunter (Senior Editor) and two reviewers, one of whom, Hong Zhang (Reviewer #1), is a member of our Board of Reviewing Editors. The following individual involved in review of your submission has agreed to reveal their identity: Qingjun Wang (Reviewer #2).

The reviewers have discussed the reviews with one another and the Reviewing Editor has drafted this decision to help you prepare a revised submission.

Summary:

In this manuscript, the authors utilized *Fundc1* whole body KO, as well as *Fundc1* and *Atg5* megakaryocyte/platelet-lineage specific KO to demonstrate that autophagy, particularly *Fundc1*-mediated mitophagy, in platelets is critical for degrading mitochondria as well as controlling mitochondrial functions and platelet activation under both normoxia and more importantly, hypoxia conditions. Moreover, the authors convincingly showed that in WT, hypoxia greatly induces platelet mitophagy, reduces platelet mitochondrial quality, and inactivates platelets; these hypoxic effects are blunted by genetic loss of *Fundc1* or a cell-penetrating, peptide inhibitor specific to *Fundc1*-mediated mitophagy. Besides revealing (patho) physiological settings for *Fundc1*-mediated mitophagy, this work is also highly significant from the translational point of views: The authors show that platelet-specific KO of *Fundc1* or *Atg5* reduces I/R injury and protects heart function in a mouse model of myocardial infarction. The authors further show that hypoxic preconditioning, which leads to platelet mitophagy, impairs ATP-producing oxidative phosphorylation and platelet activation, ameliorating I/R injury in mice in a *Fundc1*-mediated mitophagy-dependent manner. This work provides a possible novel route for preventing severe I/R injury by inducing platelet mitophagy and inactivating platelets a priori. Overall, the work is interesting. The reviewers found that additional work is required to strengthen the manuscript to meet the standards for *eLife*.

Essential revisions:

1) Lazarou et al. (2015) Nature 524:309 shows that TOM20 is primarily degraded by the ubiquitin-proteasome system rather than autophagy under the conditions they used. It will strengthen the current manuscript to examine additional mitophagy markers (e.g., COXII Western blotting). *Fundc1*KO in Figure 1 – hypoxia-induced degradation of TOM20, Tim23 and p62 was greatly but not completely inhibited. This could be due to the ubiquitin-proteasome system and/or other adapter-mediated mitophagy. Along this line, among the many mitophagy adaptors/mediators, is FUNDC1 the primary mediator in platelet mitophagy? Transcriptional regulation of hypoxia-induced NIX- or BNIP3-mediated mitophagy was briefly discussed. But other adaptors/mediators are not discussed.

2) Subsection “Hypoxia causes extensive mitophagy in vivo in a FUNDC1-dependent manner in different tissues”, second paragraph: From Figure 1, total FUNDC1 decreases upon prolonged hypoxia. It is unclear what fraction of the reduction in p18 FUNDC1 is due to dephosphorylation, and what fraction is due to loss of FUNDC1. One possible way to clarify this is a FUNDC1 IP followed by p18 western blotting. The authors can use the samples from the experiments analyzed for LC3-FUNDC1 interaction.

3) Figure 1. Quantification of levels of Tom20, Tim23 and p62 should be provided. Under normal conditions, levels of Tom20, Tim23 and p62 are higher in FUNDC1 KO platelets than those in control mice. Does this suggest that FUNDC1 is involved in mitophagy under normal conditions? The mitophagy data shown in other panels in Figure 1, Figure 3, Figure 4, Figure 5, and relevant supplemental figures should also be quantified. Saturating blots in some of these figures should be replaced with lower exposure blots. In Figure 1, quantification of mitochondria-enclosed autophagosome should be provided. It is highly likely that there are mitophagosomes in WT platelets under normoxia conditions. The authors claimed that mitophagosomes were not observed in platelets from hypoxic FUNDC1 KO mice. How are the damaged mitochondria in FUNDC1 KO platelets degraded?

---

## [Author Response]

*Essential revisions:*

*1) Lazarou et al. (2015) Nature 524:309 shows that TOM20 is primarily degraded by the ubiquitin-proteasome system rather than autophagy under the conditions they used. It will strengthen the current manuscript to examine additional mitophagy markers (e.g., COXII Western blotting). Fundc1KO in Figure 1 – hypoxia-induced degradation of TOM20, Tim23 and p62 was greatly but not completely inhibited. This could be due to the ubiquitin-proteasome system and/or other adapter-mediated mitophagy. Along this line, among the many mitophagy adaptors/mediators, is FUNDC1 the primary mediator in platelet mitophagy? Transcriptional regulation of hypoxia-induced NIX- or BNIP3-mediated mitophagy was briefly discussed. But other adaptors/mediators are not discussed.*

We have now added western blotting of COXII as a biochemical marker of mitophagy. As both the inner membrane and outer membrane mitochondrial proteins are degraded simultaneously, we suggest that mitophagy, rather than the ubiquitin-proteasome system, is responsible for these changes.

As the reviewer nicely pointed out, our results convincingly showed that mitophagy occurs in platelets in a FUNDC1-dependent manner in response to hypoxia. We do not rule out the involvement of other mediators in this process. We have now included discussion of the roles of Parkin and NIX in mitophagy in platelets (Discussion, second paragraph). We believe that different autophagy/mitophagy adapters play distinct roles under different stress conditions.

*2) Subsection “Hypoxia causes extensive mitophagy* in vivo *in a FUNDC1-dependent manner in different tissues”, second paragraph: From Figure 1, total FUNDC1 decreases upon prolonged hypoxia. It is unclear what fraction of the reduction in p18 FUNDC1 is due to dephosphorylation, and what fraction is due to loss of FUNDC1. One possible way to clarify this is a FUNDC1 IP followed by p18 western blotting. The authors can use the samples from the experiments analyzed for LC3-FUNDC1 interaction.*

We have performed the FUNDC1 IP experiment followed by p18 western blotting as you suggested (see the revised Figure 1). Our published papers (Chen G, et al. 2014, Kuang, et al. 2016) demonstrated that dephosphorylation of FUNDC1 enhances its interaction with LC3 by 5-6 fold. Our most recent data has identified an E3 ligase that is responsible for FUNDC1 degradation and addition of ubiquitin to FUNDC1 occurs before dephosphorylation (manuscript in revision for EMBO reports).

*3) Figure 1. Quantification of levels of Tom20, Tim23 and p62 should be provided. Under normal conditions, levels of Tom20, Tim23 and p62 are higher in FUNDC1 KO platelets than those in control mice. Does this suggest that FUNDC1 is involved in mitophagy under normal conditions? The mitophagy data shown in other panels in Figure 1, Figure 3, Figure 4, Figure 5, and relevant supplemental figures should also be quantified. Saturating blots in some of these figures should be replaced with lower exposure blots. In Figure 1, quantification of mitochondria-enclosed autophagosome should be provided. It is highly likely that there are mitophagosomes in WT platelets under normoxia conditions. The authors claimed that mitophagosomes were not observed in platelets from hypoxic FUNDC1 KO mice. How are the damaged mitochondria in FUNDC1 KO platelets degraded?*

We have now provided quantification of the levels of Tom20, Tim23 and P62 in all the figures (Figure 1, Figure 3, Figure 4 and Figure 5) and the supplemental figures. Under normal conditions, levels of Tom20, Tim23 and p62 are higher in FUNDC1 KO platelets than those in control mice, indicating that FUNDC1 may be involved in mitophagy under normal conditions. Some of the saturating blots in some of these figures have been replaced with lower exposure blots (see Figure 1, Figure 4, Figure 5).

We also provided the quantification of mitochondria-enclosed autophagosomes (see legends in Figure 1, Figure 1—figure supplement 3 and Figure 1—figure supplement 4). We agree with the reviewer that it is highly likely that there are mitophagosomes in WT platelets under normoxia conditions. As mitochondrial autophagy is a rapid process, it is hard to detect mitophagosomes in WT platelets under normoxia conditions.